# Highly Active Manganese Oxide from Electrolytic Manganese Anode Slime for Efficient Removal of Antibiotics Induced by Dissociation of Peroxymonosulfate

**DOI:** 10.3390/nano13101600

**Published:** 2023-05-10

**Authors:** He Zhang, Ruixue Xiong, Shijie Peng, Desheng Xu, Jun Ke

**Affiliations:** School of Chemical and Environmental Engineering, Wuhan Institute of Technology, Wuhan 430205, China2003181414@stu.wit.edu.cn (S.P.);

**Keywords:** solid waste recycling, electrolytic manganese anode slime, manganese oxide, antibiotics, advanced oxidation process

## Abstract

In this paper, high-activity manganese oxide was prepared from electrolytic manganese anode slime to realize the efficient removal of antibiotics. The effects of sulfuric acid concentration, ethanol dosage, liquid–solid ratio, leaching temperature and leaching time on the leaching of manganese from electrolytic manganese anode slime were systematically studied. Under the optimal conditions, the leaching rate of manganese reached 88.74%. In addition, a Mn_3_O_4_ catalyst was synthesized and used to activate hydrogen persulfate (PMS) to degrade tetracycline hydrochloride (TCH). The synthesized Mn_3_O_4_ was characterized by XRD, XPS, Raman, SEM and HRTEM. As a result, the prepared Mn_3_O_4_ is spherical, with high purity and crystallinity. The catalytic activity of Mn_3_O_4_ for PMS to degrade TCH was increased to 82.11%. In addition, after four cycles, the performance remained at 78.5%, showing excellent stability and recyclability. In addition, O_2_^−^ and ^1^O_2_ are the main active species in the degradation reaction. The activity of Mn_3_O_4_ is attributed to it containing Mn(II) and Mn(III) at the same time, which can quickly realize the transformation of high-valence and low-valence manganese, promote the transfer of electrons and realize the degradation of organic pollutants.

## 1. Introduction

Recently, more and more antibiotics are detected in natural waterbodies owing to the increasing abuse of antibiotics that cannot be completely metabolized by humans and animals. Usually, the antibiotics possess high hydrophilicity and biological stability, which can migrate and augment in the environment, thereby causing bacterial resistance and bioconcentration. Conventional treatment methods such as coagulation, sedimentation, adsorption and membrane separation cannot completely remove antibiotics, so it is necessary to develop an efficient and promising method to remove antibiotics in water environments.

Advanced oxidation process(AOPs) are a variety of reactions depending on strong oxidation free radicals, which can degrade antibiotic pollutants efficiently [1,2,3]. During the AOPs, two common peroxides, namely hydrogen peroxide and peroxosulfates (PDS/PMS), have been widely studied because radicals with high redox potentials, such as hydroxyl radical (•OH, 1.9–2.7 V) and sulfate radical (SO_4_•^−^, 2.5–3.1 V), can be produced efficiently [4,5]. It is demonstrated that, owing to low oxidation capacity, the peroxosulfates can be activated by various activators, such as heat [6], electricity [7], ultraviolet [8], transition metal ions [9], ultrasound [10] and heterogenous metal oxides [11]. Meanwhile, various reactive oxygen species (ROS), including SO_4_•^−^, •OH, superoxide radical (O_2_•^−^) and singlet oxygen (^1^O_2_) can be formed rapidly [12,13,14,15]. Furthermore, in the PMS/PDS-based system, it is found that a large amount of ^1^O_2_ is produced by non-free radical pathway, which can rapidly degrade/mineralize organic pollutants in water [16]. In addition, the formed ^1^O_2_ can promote the redox cycle of metals and accelerate electron transfer [17]. PMS is a representative asymmetric persulfate structure, whereas PDS is a crystal with symmetrical structure, which is highly stable. In terms of structure, activation pathway and activation difficulty, PMS is superior to PDS in overall performance and is easier to activate [18]. However, the activation of PMS by heat, ultraviolet rays and ultrasonic waves faces problems such as high energy consumption, poor treatment effect and high equipment cost. At the same time, compared with the homogeneous catalytic system composed of transition metal ions such as Co^2+^, Fe^2+^ and Ag^+^, the heterogeneous catalytic activation method of transition metal oxides has the advantages of convenient catalyst recovery and less secondary pollution [19,20,21]. Therefore, it is urgent to look for a heterogeneous system for deep oxidation to produce a large amount of ^1^O_2_. Recently, it has been proved that multivalent metal oxide, Mn_3_O_4_ possessing two valent Mn of Mn(II) and Mn(III), can be used as a catalyst to activate PMS and presents an excellent degradation efficiency. For example, Reza Shokoohi et al. successfully prepared Mn_3_O_4_ nanoparticles with a chemical precipitation method for activating PMS to degrade reactive blue 222 [22]. In addition, Syaifullah Muhammad et al. prepared Mn_3_O_4_ with a hydrothermal method to rapidly activate PMS to degrade phenol [23].

Electrolytic manganese anode slime (EMAS) is a solid waste produced in the process of producing electrolytic manganese metal, which contains many impurities and harmful substances such as selenium, manganese and ammonia nitrogen. Traditional treatment methods make harmful substances enter surface water and groundwater with leachate [24], which will seriously endanger the ecological environment and human health. Meanwhile, the EMAS contains a large proportion of Mn, Fe and Al elements, which can be recycled and partially substituted for the usage of natural manganese ore [25]. For example, Zhao et al. used hydrogen peroxide as reducing agent to wet leach manganese ion from electrolytic manganese anode slime [26]. Furthermore, Yong Yang et al. recovered manganese oxide from EMAS by using a vacuum carbothermal reduction method [27]. Therefore, EMAS can be used as a raw material to prepare manganese-based catalysts and combine advanced oxidation technology to degrade organic matter. For instance, Zhao et al. used electrolytic manganese anode mud as a raw material to prepare a manganese-based catalyst by calcination route, which displayed a good performance of activating PMS to degrade tetrachlorophenol [28].

Based on the above-mentioned strategies, in this work, a highly active Mn_3_O_4_ nanocatalyst was successfully synthesized via a facile acid leaching precipitation oxidation method for reutilizing electrolytic manganese anode slime as a raw material. The recycling efficiency of Mn elements from the EMAS were systematically investigated. In addition, characterize the physical and chemical properties of Mn_3_O_4_ nano-catalyst, and the activity of the Mn_3_O_4_ nanocatalyst for activating PMS to mineralize TCH was studied. Furthermore, the mechanism of the Mn_3_O_4_ nanocatalyst for efficiently activating PMS was proposed. In this work, industrial solid EMAS was used to prepare the catalyst instead of analytical pure medicine. The prepared catalyst has good degradation performance and excellent stability in the process of treating antibiotics in water, which embodies the ideal mode of treating waste with waste.

## 2. Materials and Methods

### 2.1. Chemicals

Electrolytic manganese anode slime is a waste residue produced from an electrolytic manganese enterprise at Wuhan. Sulfuric acid (HClO_4_, AR), ethanol (C_2_H_5_OH, AR), hydrogen peroxide (H_2_O_2_, AR), sodium hydroxide (NaOH, AR) and nitric acid (HNO_3_, AR) were obtained from Shanghai Aladdin Biochemical Technology Co., Ltd., Shanghai, China. All chemicals and reagents were of analytical grade and used directly without further purification. The water used in the synthesis process and the preparation of the standard solution was ion-exchanged water.

### 2.2. Pretreatment of Electrolytic Manganese Anode Slime

A certain amount of electrolytic manganese anode slime was weighed and washed with water to remove some soluble impurities. Then, the solid was dried at 100 °C for 8 h to constant weight, finely ground and passed through a 100-mesh quasi-sieve as the raw materials.

### 2.3. Acid Leaching

Firstly, ethanol and sulfuric acid were mixed according to a set ratio to prepare a leaching agent. Then, a certain volume of the leaching agent was added into a 100 mL three-necked flask, and then heated to a set temperature in an oil bath pot. Subsequently, 5.00 g of the pretreated electrolytic manganese anode slime was poured into the three-necked flask for stirring and leaching for a certain time. When the reaction was finished, the filtrate was filtered when it was hot to obtain manganese-contained solution. Then the volume of filtrate was fixed to 250 mL, and 5 mL of leaching solution was taken to determine the manganese content. The total manganese content in the sample and leaching solution was analyzed by perchloric acid oxidation ammonium ferrous sulfate titration method. Moreover, the effects of leaching temperature, sulfuric acid concentration, ethanol dosage, liquid–solid ratio and time on the leaching efficiency were also investigated in turn, according to the single factor experiment principle.

### 2.4. Preparation of Mn_3_O_4_

The preparation procedure of the Mn_3_O_4_ nanocatalyst is displayed in Figure 1. Through acid leaching filtration, the obtained suspension was separated into leaching residue and acid leaching solution containing a large number of metal elements. Compared with liquid–liquid extraction, precipitation is a simpler and more economical method for separating metals. Therefore, for the obtained acid leaching solution, 2 mol/L NaOH solution was added dropwise to adjust the pH to 5.0 ± 0.1, and Fe and Al were removed by forming hydroxide precipitation. The solid and liquid phases were separated by centrifugation to obtain 1# filtrate. The Mn_3_O_4_ was synthesized in a 250 mL high-density polyethylene flask (the volume of 1# filtrate was 100 mL), which was placed in a water bath at 25 °C and continuously stirred (300 r/min) for 60 minutes. The pH of 1# filtrate was slowly raised to 10.0 ± 0.1 with 2 mol/L NaOH solution to obtain slurry containing Mn(OH)_2_, and then 0%, 0.1%, 0.2% and 0.4% H_2_O_2_ were added (volume ratio to 1# filtrate) to the slurry, respectively. After the complete reaction, each was centrifuged, washed with deionized water and ethanol alternately for three times, and the washed solid was dried in an oven at 70 °C for 4 h. The product was ground into powder by agate mortar and nanosized Mn_3_O_4_ powder was obtained. The prepared samples were labeled as Mn_3_O_4_-0%, Mn_3_O_4_-0.1%, Mn_3_O_4_-0.2% and Mn_3_O_4_-0.4%, respectively.

### 2.5. Characterizations

The phase of the sample was analyzed by X-ray diffractometer (XRD, X’Pert PRO MPD, Nalytical company Netherlands) with Cu Kα radiation (λ = 1.5418 Å) in the range of 10~90 degrees. The morphology and microstructure of the samples were characterized by scanning electron microscopy (SEM, Hitachi Regulus8100, HITACHI Japan) and transmission electron microscope (TEM, FEI Tecnai F20, FEI company American). By analyzing the scanned images, the microscopic morphology, pore size and aggregation degree of the sample materials can be determined. In order to determine the chemical state of elements, X-ray photoelectron spectroscopy (XPS, Thermo Kalpha, Thermo Fisher Scientific, American) was used. The degree of mineralization in PMS/Mn_3_O_4_ systems was evaluated by TOC analyzer (TOC-L). In order to determine the molecular structure of catalytic materials, a Raman spectrometer (Thermo DXR, Thermo Fisher Scientific, American, excitation wavelength = 633 nm) was used to test and analyze.

### 2.6. Activity Test

The catalytic activity of the catalyst was assessed by activating PMS to degrade tetracycline hydrochloride (TCH) in an aqueous solution. A 50 mL solution of tetracycline at 50 mg/L was added to the beaker, then 20 mg of catalyst was added to the beaker and stirred thoroughly for 30 min to reach adsorption–desorption equilibrium. This was followed by the addition of 20 mg of PMS to initiate the catalytic oxidative degradation of tetracycline, with continuous stirring during the catalytic oxidation to ensure a homogeneous mixture of catalyst, tetracycline solution and PMS. Using a syringe, 1.5 mL of the reaction solution was taken at specific time intervals and passed through an aqueous filter membrane of 0.22 μm. The resulting filtrate was quickly injected into a test tube containing 1.5 mL of ethanol solution to scavenge the free radicals.

The concentration of tetracycline was analyzed using a UV-Vis spectrophotometer. The absorbance of the supernatant was measured with a UV-Vis spectrophotometer at a maximum absorption wavelength of 356 nm. Degradation rates of organic pollutants were calculated according to the Beer–Lambert law,
$$Degradationrate\left(\%\right)=\left(\frac{A_{0}-A}{A_{0}}\right)\times 100\%=\left(\frac{C_{0}-C}{C_{0}}\right)\times 100\%$$
where A_0_ and A are the initial absorbance and the absorbance of antibiotics at time t. C_0_ and C are the initial concentration of TCH and the concentration at time t.

A systematic single factor conditions the experiments: the other conditions were set to be the same and one of the process parameters’ conditions was changed, each in turn, to study the catalyst dosage, PMS dosage, catalytic temperature and other effects on the degradation experiments.

For catalyst recoverability tests, the used solid catalyst material was recovered by filtration, washed alternately with deionized water and ethanol and dried in an oven at 70 °C for 4 h. Several parallel experiments were carried out for each run prior to the previous run to ensure that sufficient catalyst was available for the next repeat experiment.

## 3. Results and Discussion

As can be seen from Figure 1a, the leaching rate of manganese gradually increased and then stabilized as the temperature increased. The leaching rate of manganese gradually increased from 30 °C to 70 °C, and did not change significantly when the temperature was increased from 70 °C. This is because increasing the temperature accelerates the thermal movement of the ethanol molecules and facilitates the diffusion of the ethanol molecules into the electrolytic manganese slag, making the reaction more complete. At the same time, because of the low boiling point of ethanol, too high a temperature will make the ethanol molecules volatilize, which is not conducive to the reaction. Therefore, the best leaching temperature should be 70 °C, and the leaching rate of the manganese element can reach 76.42% at this temperature. As the shown in Figure 1b, the leaching rate of the manganese element increases with the increase of sulfuric acid concentration, and then decreases after reaching the highest value. Increasing the sulfuric acid concentration is beneficial to the leaching of electrolytic manganese slag, and after the peak leaching rate of the manganese element the leaching rate starts to decrease, probably because the viscosity of the system increases with the increase of sulfuric acid concentration, and the mass transfer effect becomes poor, making the system reaction insufficient. Therefore, the optimum sulfuric acid leaching concentration was 5 mol/L, and the leaching rate of manganese was 88.74% at this condition.

It can be observed from Figure 1c that the leaching of elemental manganese was the first to increase and then decrease as the volume fraction of ethanol increased. This may be due to as the ethanol concentration increases, the leaching rate increases, and the solubility of manganese in the sulfuric-ethanol-aqueous system formed by excess ethanol decreases, resulting in a decrease in the leaching rate of manganese. The presence of a large amount of unreacted ethanol in the system leads to an increase in production costs. In summary, the optimum leaching condition was a 5% volume fraction of ethanol, which resulted in a leaching rate of 88.74% of manganese. As can be seen in Figure 1d, the leaching rate of manganese gradually increases with the gradual increase of the liquid to solid ratio and then stabilizes. This is because, with the increase of the liquid to solid ratio, the leaching dose of the system is large and the mass transfer effect is good, which is conducive to the full occurrence of the reaction, but too high a liquid to solid ratio will affect the processing capacity and cost of the equipment. In summary, the optimum liquid to solid ratio was six, and the leaching rate of manganese element under this condition was 88.74%. As can be seen from Figure 1e, the leaching of manganese was the first to increase and then stabilize as the reaction time increased. This reflects that the electrolytic manganese slag has good oxidizing properties and is able to oxidize ethanol quickly. The optimum reaction time is 4 h, and the leaching rate of manganese under these conditions is 88.74%.

Figure 2a is the XRD pattern of Mn_3_O_4_ synthesized at different H_2_O_2_ doses. The red PDF#18-0404 card is the standard card of MnO(OH), and the black PDF#24-0734 card is the standard card of Mn_3_O_4_. Among the four materials with different amounts of hydrogen peroxide, the peak of Mn_3_O_4_-0.2% is the sharpest, indicating that the crystallinity is the highest. As shown in Figure 2a, the Mn_3_O_4_-0% material without added hydrogen peroxide shows the peak shapes of Mn_3_O_4_ (103) and (211) crystal faces at 33 and 36, respectively, which may be because when Mn(OH)_2_ precipitates, the surface layer Mn(OH)_2_ reacts with oxygen in the air to generate MnO(OH), and MnO(OH) is drying. However, Mn(OH)_2_ wrapped in it was dried to remove H_2_O to form MnO, and MnO, Mn_2_O_3_ and MnO together showed the peak of Mn_3_O_4_. Under these conditions, the peak of Mn_3_O_4_ is the most gentle and the crystallinity is the worst. The Mn_3_O_4_ produced by adding 0.1% hydrogen peroxide has obvious peaks at 25, 30 and 34. The peak at 25 matches the PDF#18-0404 card, showing the peak of MnO(OH), while the peaks at 30 and 34 show the peak of MnOx, indicating that Mn_3_O_4_ under this condition may have different states due to incomplete oxidation. With the increase of the dosage of hydrogen peroxide, the X-ray diffraction pattern of the material shows a peak represented by Mn_3_O_4_, and the XRD patterns of Mn_3_O_4_-0.2%, Mn_3_O_4_-0.4% and Mn_3_O_4_-0.8% are highly consistent, which may be because hydrogen peroxide is enough to partially oxidize Mn(OH)_2_ to generate Mn_3_O_4_. The planar diffraction peak is completely matched with the tetragonal Mn_3_O_4_ (hausmannite, JCPDS NO.24-0734) [29], and the 2θ diffraction corresponds to 18, 29, 32, 36, 38, 45, 51, 54, 56, 58 and 55. In addition, XRD patterns show a few other crystallization peaks, which may be due to the fact that the raw material for preparation is electrolytic manganese anode mud, which is solid waste and contains many impurities, so the purity and crystallinity of prepared Mn_3_O_4_ are slightly worse than those prepared by chemical reagents. As shown in the previous study [30], it is found that the catalyst with higher crystallinity and purity can activate PMS more effectively and achieve greater removal efficiency. Figure 2b shows the Raman spectra of Mn_3_O_4_-0.2% nanostructures before the reaction, which reveals two strong phonon vibration modes of tetragonal Mn_3_O_4_ phase nanostructures. The main peak observed at 643 cm^−1^ can be attributed to the A1g vibration mode caused by respiratory vibration of Mn2p ions in a tetragonal Mn_3_O_4_ phase. The small peak observed at 359 cm^−1^ can be attributed to the T2g vibration mode caused by bending vibration [31,32], which is the characteristic peak produced by Mn_3_O_4_ crystal structure. This is consistent with the previously reported results of Mn_3_O_4_ [33,34]. Therefore, it is further indicated that the synthesized catalyst is Mn_3_O_4_.

The morphology and structure of Mn_3_O_4_ was studied using scanning electron microscopy (SEM) and the images are shown in Figure 3a,b. The Mn_3_O_4_ obtained by precipitation oxidation was a three-dimensional nanoparticle, and a spherical structure with a regular and smooth surface, a relatively uniform particle size and good dispersion, with a particle size around 100 nm, could be observed. This is consistent with the previously reported Mn_3_O_4_ nanocatalysts [35,36] and the formation of such a microsphere structure greatly increased the surface area of Mn_3_O_4_, which can provide more contact for activating PMS sites. Through observation with a high-resolution transmission electron microscope (HRTEM), it was further confirmed that the phase of Mn_3_O_4_ material is as shown in Figure 3c,d. Clear lattice stripes can be plainly observed in the figure, which indicates that Mn_3_O_4_ nanoparticles are highly crystalline. The lattice stripes with lattice distances of about 0.493, 0.272 and 0.249 nm correspond to the (101), (103) and (211) planes of tetragonal <Mn_3_O_4_-0.2%>, which are consistent with those of XRD analysis.

The elemental composition and valence distribution of the prepared catalyst was determined by X-ray photoelectron spectroscopy (XPS) and the results are shown in Figure 4. The O 1s and Mn 2p spectra were clearly observed from the XPS survey scans (Figure 4a), from which no other elements were found apart from indeterminate carbon.

The survey scan shows the catalyst to be a manganese oxide. As shown in Figure 5a, the high-resolution Mn 2p spectrum can be split into two or three components by fitting analysis to the XPS peaks of the Gaussian distribution. In the XPS spectrum of Mn 2p, the signal peaks at 654.3 eV, 652.7 eV, 643.2 eV and 641.3 eV are attributed to Mn^2+^ 2p_1/2_, Mn^3+^ 2p_1/2_, Mn^2+^ 2p_3/2_ and Mn^3+^ 2p_3/2_ in the oxide, respectively (Figure 4b), which indicates that the chemical state of the element Mn in this manganese oxide has two valence states, Mn^3+^ and Mn^2+^. Similar results have been found in previous literature [29,37]. It is worth noting that the ratio of Mn^2+^ to Mn^3+^ is calculated from the corresponding peak area to be about 1:2, which further indicates that the prepared catalyst is Mn_3_O_4_ [29].

In order to investigate the optimal degradation conditions of the prepared Mn_3_O_4_ catalysts for TCH, we conducted systematic single-factor condition experiments and obtained the catalytic performance of the Mn_3_O_4_ nanocatalysts as shown in Figure 5. As shown in Figure 5a, for the TCH solution with the addition of PMS only, the degradation rate of TCH was only 29.04%, but the rate and efficiency of TCH degradation was significantly increased with the addition of the Mn_3_O_4_ catalyst. After 30 min of catalytic degradation under simulated organic wastewater conditions, the Mn_3_O_4_ sample prepared with activated PMS could reach 82.11% TCH degradation efficiency, and within five minutes of adding PMS, the TCH removal rate could reach 70%, which was significantly higher compared to the addition of PMS only. The increased efficiency of TCH degradation is attributed to the fact that Mn_3_O_4_ can provide the catalytic site (Mn) to activate PMS for TCH degradation.

The reaction temperature as in Figure 5b and the amount of PMS as in Figure 5c were explored and it can be found that, within the scope of 25 °C~35 °C, the temperature has no more obvious effect on the degradation reaction of TCH, and a higher temperature will cause the thermal decomposition of PMS. It is impossible to explore the effect of the prepared Mn_3_O_4_ nanocatalyst on the activation of PMS, so the higher temperature was not explored. With the increase of PMS usage, the removal rate of TCH increased from 79.02% to 83.8%. It could be found that although increasing the amount of PMS could increase the PMS removal rate, the increase was only 4.78%, and the gain effect was not obvious. This may be because when the PMS dosage was 10 mg, the catalytic reaction had not yet reached saturation and an increase in the PMS dosage would lead to an increase in radical generation, and therefore an increase in the TCH removal rate. When the dosage of PMS is increased again, excessive free radicals will produce antagonism leading to increased consumption, so the gain is not obvious.

By the degradation effect of TCH solution under simulated experimental conditions, the catalytic abilities of Mn_3_O_4_-0%, Mn_3_O_4_-0.1%, Mn_3_O_4_-0.2% and Mn_3_O_4_-0.4% were studied. The adsorption-desorption equilibrium was achieved by stirring for 30 min. As shown in Figure 5d, for the degradation of organic wastewater, with the increase of hydrogen peroxide dosage, the adsorption capacity decreased moderately. Under the condition of simulated wastewater, after 30 minutes, the prepared Mn_3_O_4_-0% sample could only degrade 58.4% of TCH. Although Mn_3_O_4_-0.1% has good catalytic activity and the degradation rate can reach 64.7%, the catalytic activity was still not up to expectations. Compared with Mn_3_O_4_-0% and Mn_3_O_4_-0.1%, the catalytic activity of the Mn_3_O_4_-0.2% material was significantly improved, and the degradation efficiency of TCH reached 82.11% within 30 minutes. Although the degradation efficiency of Mn_3_O_4_-0.1% is lower than that of Mn_3_O_4_-0.2%, compared with the sample without added hydrogen peroxide the degradation efficiency is obviously improved. Increasing the dosage of hydrogen peroxide did not significantly improve degradation effect of TCH because the Mn^2+^ in the solution had reached oxidation saturation to form Mn_3_O_4_. The improvement of the Mn_3_O_4_-0.2% catalytic activity is attributed to the presence of Mn^2+^and Mn^3+^ at the same time, which can quickly transform from low-valence manganese to high-valence manganese and effectively stimulate more free radicals to degrade TCH.

The stability and reusability of catalysts are important indexes to evaluate their practical application performance. Figure 6a shows the degradation performance of TCH with Mn_3_O_4_-0.2% for four cycles. After four cycles, the removal efficiency can still reach 78.5%, which is only 3.6% lower than the first removal efficiency. The probable reasons can be generalized as two aspects: (1) the slight leaching of Mn ions in the material may lose some active sites of the catalyst; (2) the residual TCH and its by-products on the catalyst will compete with TCH for active sites, resulting in poor removal efficiency. The test shows that the material has excellent stability. Raman spectra of the original and used samples of Mn_3_O_4_-0.2% are shown in Figure 6b,c, which demonstrates that the crystal structure of the catalyst did not change after four cycles. The results show that Mn_3_O_4_-0.2% has excellent constancy and reusability, which suggests that it has a broad application prospect in organic wastewater treatment.

In order to further explore the catalytic activity of the Mn_3_O_4_-0.2% nanocatalyst developed in this work, we used this catalyst to degrade actual medical wastewater, and analyzed the COD value in the wastewater after the reaction by using the permanganate index determination method described in GB 11892-89. As shown in Figure 7, under the conditions of 25 °C, catalyst dosage of 8 g/L and PMS dosage of 8 g/L, the COD removal rate can reach 43.27% within 60 min, showing certain catalytic activity. In the blank experiment, the removal rate of COD reached 33.07% at 60 min and 34.76% at 90 min. The data show that the catalyst improves reaction rate and removal efficiency. In addition, the ions in medical wastewater may provide electrons to PMS, promote its activation and increase its COD removal effect. To sum up, this material has a definite application prospect in the front-end pretreatment of actual wastewater.

In addition, Table 1 shows a comparison of the catalytic performance of the manganese-based catalysts prepared in this work and other reported manganese-based catalysts/PMS systems. It can be seen that Mn_3_O_4_-0.2% has reached a level similar to other reports in this work, but the degradation time of Mn_3_O_4_-0.2% is shorter, the concentration of degradable pollutants is higher, the cycle performance is better and it shows higher stability.

To explore the catalytic mechanism of Mn_3_O_4_-0.2%, The main active substances in the catalytic degradation process were identified by radical capture experiments. According to previous studies [30], there may be free radical such as SO_4_, HO and ^1^O_2_ in the system. The effects of tert-butanol (TBA), methanol (MeOH), L-histidine and p-benzoquinone (p-BQ) on TCH removal were investigated under the conditions of TCH mass concentration of 50 mg/L, PMS concentration of 0.4 g/L, pH = 7.5 and temperature of 25 °C. Methanol has high reactivity with SO_4_^−^ and OH, and can quench these two free radicals at the same time [43], while tert-butyl alcohol can only quench OH [44]. P-benzoquinone and L- histidine are used as quenchers for O_2_^−^ and ^1^O_2_, respectively [45,46]. When no capture agent is added, the degradation efficiency of Mn_3_O_4_-0.2% is 82.1%. As shown in Figure 8, when tert-butyl alcohol was added to the reaction system of Mn_3_O_4_-0.2%/PMS, the removal rate of TCH hardly changed after 30 min, which indicated that OH was not a reactive species affecting TCH degradation. When methanol was added to the system, the removal rate of TCH did not change after 30 min, indicating that SO_4_- was not a reactive species that affected TCH degradation. Under the same conditions, the removal rate was 24.2% after adding L- histidine, and 45.1% after 30 min when adding p-benzoquinone to the system. Therefore, in the process of Mn_3_O_4_-0.2%/PMS treatment of TCH, after adding L-histidine, the degradation of TCH was seriously inhibited, and its removal rate could only reach 24.2%. Moreover, the degradation of TCH reached a balance after 5 min of treatment, which indicated that most of the active species generated by Mn_3_O_4_-0.2% activated PMS and were eliminated by L- histidine, and O_2_^−^ and ^1^O_2_ existed simultaneously in the system. The above quenching experiment results show that in the process of treating TCH with Mn_3_O_4_-0.2%/PMS, the contribution of ^1^O_2_ is relatively greater, which plays a leading role in oxidation.

The possible activation mechanism of Mn_3_O_4_/PMS is shown in Figure 9. In the process of PMS activation, PMS needs to be adsorbed on the surface of the catalyst first and then the corresponding oxidizing species are generated through the fracture of PMS. When PMS comes into contact with the active site on the surface of Mn_3_O_4_, it transfers electrons through a reduction–oxidation reaction. Firstly, surface hydroxylated Mn(III) and Mn(II) provide electrons to PMS as active sites, which promotes the production of SO_4_•^−^ and •OH. At the same time, PMS can provide an electron to Mn(III) to generate SO_5_•^−^, and Mn(III) can be converted into Mn(II). SO4 •^−^ could react with H_2_O and OH^−^ to generate •OH. O2•^−^ is generated as formula (6) through the decomposition of HO_2_•^−^, or an electron is obtained through O_2_ to form [47,48,49]. PMS can generate ^1^O_2_ by self-decomposition (Equation (1)) [17]. At the same time, ^1^O_2_ can also be generated by the Equations (2)–(4) [47,50]. According to the experimental results, with the production of O_vac_, O_latt_ is released and transformed into reactive oxygen species(O*), which can further react with PMS and finally be converted into ^1^O_2_ (Equations (7)–(9)) [48,49,50,51]. The redox of Mnn+/Mn(n+1)+ in the lattice is promoted by the generation of O_vac_ [17]. SO_4_•^−^ and •OH are produced in the intermediate reaction, but in the Mn_3_O_4_/PMS system of this experiment, these two free radicals did not make a significant contribution, and they mainly played the role of free radicals O_2_^−^ and ^1^O_2_, which degraded TCH.
HSO_5_^−^ + SO_5_^2−^ → HSO_4_^−^ + SO_4_^2−^ + ^1^O_2_(1)
2•OH → 1/2^1^O_2_ + H_2_O(2)
O_2_•^−^ + •OH → ^1^O_2_ + OH^−^(3)
2O_2_•^−^ + 2H^+^ → ^1^O_2_ + H_2_O_2_(4)
SO_5_^2−^ + H_2_O → HO_2_ •^−^+ SO_4_^2−^ + H^+^(5)
SO_5_^2−^ + HO_2_ •^−^ → SO_4_^2−^ + H^+^ + O_2_•^−^(6)
≡Mn^(n+1)+^ + O_2_^−^(O_latt_) → ≡Mn^(n)+^ + O_vac_ + 0.5O_2_(7)
O_vac_ → O*(8)
O* + HSO_5_^−^ → HSO_4_^−^ + ^1^O_2_(9)

## 4. Conclusions

In summary, the Mn_3_O_4_ catalyst is successfully prepared using precipitation–oxidation method. Meanwhile, the optimum leaching conditions of electrolytic manganese slag were explored. The optimum leaching conditions of manganese were 70 °C, 5 mol/L sulfuric acid concentration, 1:6 g/L liquid-solid ratio, 5% ethanol volume fraction, 4 h acid leaching time and 88.74% leaching rate. The optimum conditions for degradation of TCH by Mn_3_O_4_ nanocatalyst were investigated. Considering the catalytic activity and economic cost, the dosage of catalyst was 0.4 g/L, the reaction temperature was 25 °C and the dosage of PMS was 0.4 g/L. It was found that Mn_3_O_4_ has a smooth surface and uniform particle size. In addition, the free radical capture experiment shows that O_2_^−^and ^1^O_2_ are the main active species in the degradation reaction. As a result of Mn(II) and Mn(III) acting at the same time, it has a good activation effect on PMS and Mn_3_O_4_ enhances the catalytic degradation activity of the environmental organic pollutants. At the same time, compared with pure Mn_3_O_4_, the cost of synthesizing Mn_3_O_4_ from electrolytic manganese slag is lower, and the synthesis method is green and environmentally friendly. This study proposes a new idea for the effective utilization of electrolytic manganese slag and realizes the new goal of waste treatment and green recycling development.

## Data Availability

The data presented in this study are available on request from the corresponding author.

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
