# Peer review of "Highly Active Manganese Oxide from Electrolytic Manganese Anode Slime for Efficient Removal of Antibiotics Induced by Dissociation of Peroxymonosulfate"

_nanomaterials, 2023, doi:10.3390/nano13101600_

Round 1

Reviewer 1 Report

In this paper Mn3O4 catalyst was synthesized by precipitation-oxidation method, and the optimum leaching conditions of electrolytic manganese slag were explored. The authors have reported only few data concerning the characterization of the prepared materials. XRD, Raman, XPS discusion must be properly improved. The quality of XRD spectra is very poor. The commercial Mn3O4 reference must be properly characterized. 

See Fig 2b: "Raman spectrum before the reaction": please add specific comments in the label in order to clarify the catalyst and the reaction conditions. 

In the manuscript some sentences must be revised

Reviewer 2 Report

 The submitted manuscript describes the procedure to valorize electrolytic manganese anode slime as a catalyst for the advanced oxidation treatment of antibiotics. The paper is well-written, and the presented research is sufficient to prove the concept of this work. There only few points that authors should improve before publication. What I would like to express is that the subject of this paper is better fitted to the field of circular economy, waste treatment engineering and chemical engineering rather than a nanomaterial development. Therefore, I am not sure if the journal suits this paper.

The abstract does not mention that the application of this work is for advanced characterization processes and that is not clear from the beginning to the reader.

Scheme 1 should mention the solids separated in each step. For instance the Fe/Al.

English usage should be upgraded

Reviewer 3 Report

MS No: 

nanomaterials-2370133

Title:

Highly active manganese oxide from electrolytic manganese anode slime for efficient removal of antibiotics induced by dissociation of peroxymonosulfate

Authors:     

He Zhang, Ruixue Xiong, Shijie Peng, Desheng Xu and Jun Ke

The present manuscript deals with the synthesis of manganese oxide from electrolytic manganese anode slime and their activity towards tetracycline hydrochloride degradation in water induced by peroxymonosulfate activation. In my opinion, it can be accepted for publication in Nanomaterials after minor revision.

·       Figure 5. The catalyst and PMS dosage should be expressed as concentration (eg mg/L, mM).

·       More information concerning the initial and final pH of degradation experiments should be added.

·       I would advise the authors to test the activity of the present system under more realistic conditions such as real water matrix (wastewater, bottled water)

English quality is fine

Reviewer 4 Report

The authors Zhang et al, reported their work on titled, Highly active manganese oxide from electrolytic manganese anode slime for efficient removal of antibiotics induced by dissociation of peroxymonosulfate. Although, this work contains some results, the organization and interpretation of the result should be enhanced further. Hence, I recommend this work required a substantial revision before considering for publications.

1.      Provide the obtained results in the abstract in more concise.

2.      In section 2.1, provide the chemical purity.

3.      In Figure 2, author mentioned 2 PDF number which is belong to what?. One is red and other one is black colored PDF, this to be noted.

4.      In Fig 3c and d, index the material name belong to that d-spacing value.

5.      Novelty of the work should be highlighted in the introduction in more clearly.

6.      The authors designed the work nicely, merely presented the results but failed to discuss the observed results elaborately.

7.      I suggest the authors to compare the previous literature similar to that work to find a merits of this work.

8.      Refer and include the following references to strengthen the current version of the draft; Renewable & Sustainable Energy Reviews, 143 (2021) 110849; Current Opinion in Solid State & Materials Science 24 (2020) 100805.

The authors Zhang et al, reported their work on titled, Highly active manganese oxide from electrolytic manganese anode slime for efficient removal of antibiotics induced by dissociation of peroxymonosulfate. Although, this work contains some results, the organization and interpretation of the result should be enhanced further. Hence, I recommend this work required a substantial revision before considering for publications.

1.      Provide the obtained results in the abstract in more concise.

2.      In section 2.1, provide the chemical purity.

3.      In Figure 2, author mentioned 2 PDF number which is belong to what?. One is red and other one is black colored PDF, this to be noted.

4.      In Fig 3c and d, index the material name belong to that d-spacing value.

5.      Novelty of the work should be highlighted in the introduction in more clearly.

6.      The authors designed the work nicely, merely presented the results but failed to discuss the observed results elaborately.

7.      I suggest the authors to compare the previous literature similar to that work to find a merits of this work.

8.      Refer and include the following references to strengthen the current version of the draft; Renewable & Sustainable Energy Reviews, 143 (2021) 110849; Current Opinion in Solid State & Materials Science 24 (2020) 100805.

Round 2

Reviewer 1 Report

The current manuscript has been adequately revised ad can be accepted

Reviewer 4 Report

The revised version improved well and it may be considered for publication. 

The revised version improved well and it may be considered for publication.